# UFO-RL: Uncertainty-Focused Optimization for Efficient Reinforcement Learning Data Selection

**Yang Zhao♠, Kai Xiong♠,Xiao Ding♠†,Li Du ♡,YangouOuyang♠,Zhouhao Sun♠,**
Jiannan Guan♠,Wenbin Zhang♣, Bin Liu♣,Dong Hu♣, Ting Liu ♠and Bing Qin♠
♠Research Center for Social Computing and Interactive Robotics
Harbin Institute of Technology, China
♡Beijing Academy of Artificial Intelligence, Beijing, China
♣Du Xiaoman Technology (Beijing) Co., Ltd.
{yangzhao, kxiong, xding, oyyo, hzsun, jnguan, tliu, qinb}@ir.hit.edu.cn
duli@baai.ac.cn
zhangwenbin,liubin,hudong@duxiaoman.com

## Abstract

A primary impediment to scaling reinforcement learning (RL) for large language model (LLM) training is the substantial computational cost, predominantly arising from the necessity of multi-sampling for policy optimization and evaluation. This underscores the critical yet challenging nature of efficient training data selection. Drawing inspiration from the Zone of Proximal Development (ZPD) theory, which posits that learners acquire knowledge more effectively from tasks of intermediate difficulty, we hypothesize that LLMs exhibit optimal learning from data they have not yet mastered but demonstrate the potential to comprehend. Conventional methodologies for assessing data difficulty or informativeness typically rely on computationally intensive multi-sampling or iterative procedures. To address this limitation, we introduce UFO-RL (**U**ncertainty-**F**ocused **O**ptimization for **R**einforcement **L**earning), a novel framework that employs a computationally efficient single-pass uncertainty estimation technique to identify informative training instances. This method, requiring only a single forward pass per instance, thereby obviating the need for repeated sampling runs, achieves a significant acceleration (up to $185\times$) in data evaluation compared to multi-sampling approaches. UFO-RL leverages this efficient metric to select data within the model's estimated ZPD for training. Extensive experimentation across diverse LLMs and mathematical benchmarks demonstrates that training with a mere 10% of the data, carefully selected by UFO-RL, yields performance comparable to or even surpassing that of full-data training. Furthermore, this targeted data selection results in up to a $16\times$ reduction in overall training time, concurrently enhancing training stability and improving generalization capabilities. Thus, UFO-RL presents a practical and highly efficient strategy for scaling RL fine-tuning of LLMs by focusing learning efforts on the most informative and valuable data, thereby mitigating the computational bottlenecks associated with traditional RL training.The source code for this work is publicly available at: https://github.com/zy125413/UFO_RL.

## 1 Introduction

RL has emerged as a powerful paradigm for fine-tuning LLMs, enabling them to acquire advanced capabilities and align their outputs with desired behaviors [18, 6, 4, 14, 28]. By optimizing decision-

---

†Corresponding Author.

39th Conference on Neural Information Processing Systems (NeurIPS 2025).

making strategies based on reward signals, RL equips LLMs with complex reasoning abilities, especially for challenging tasks like mathematical problem-solving [16, 17, 4, 6].

Despite its immense promise, applying RL to LLMs, especially for complex reasoning tasks, faces significant challenges. Primary among these is the high computational cost from multi-sampling per instance for robust policy gradient estimation and evaluation [23, 2, 13], creating a major training bottleneck. Additionally, many current state-of-the-art RL methods for reasoning, such as those using policy gradient with outcome-based rewards, process data uniformly, and lack mechanisms to prioritize instances yielding the most impactful learning signal [16]. Effective training data selection is thus crucial for optimizing RL efficiency, minimizing redundant computation by focusing on high-value samples.

Existing work addresses the data efficiency issue. For instance, DAPO [23] explores removing data points with consistently correct or incorrect outcomes across multiple samplings, filtering out seemingly "easy" or "impossible" tasks. While this approach reduces computational waste associated with samples providing little learning gradient, it suffers from two main limitations. First, the identification of consistently successful or unsuccessful samples still requires the costly multi-sampling process it seeks to mitigate. Second, as data exhibiting such extreme consistency constitutes only a small proportion, this filtering technique only removes a limited amount of data, leaving the majority, where learning is most needed, in the training set, thus offering limited efficiency gains. These limitations highlight the need for a more efficient and comprehensive data selection strategy that can effectively prioritize learning opportunities across a broader range of data.

To address the limitations of current RL data selection, we draw inspiration from ZPD theory [15]. ZPD posits that optimal learning occurs when a learner engages with tasks challenging but achievable with guidance, rather than tasks that are either trivial or insurmountable. Applied to LLMs and RL fine-tuning, we hypothesize that the most informative training samples for RL are those within the model's current ZPD – tasks it has not yet fully mastered but where it shows potential for improvement. These correspond to what we term "fuzzy data", where the model's understanding or execution is incomplete or uncertain. To empirically investigate the relevance of the ZPD concept in LLM RL, we initially measure data difficulty using the accuracy observed across multiple model samplings. Our findings reveal a non-monotonic relationship between estimated data difficulty and learning effectiveness: training on overly simple data yields diminishing returns, while attempting tasks that are too complex can lead to training instability and suboptimal performance. Optimal learning occurs on data of intermediate difficulty.

While our preliminary study used multi-sampling accuracy (derived from average reward) as a difficulty proxy, this approach faces significant drawbacks. The high computational cost and the discreteness of rule-based reward signals result in a coarse granularity for data difficulty assessment, which hinders its effective practical application. To overcome these, we propose a novel, efficient uncertainty evaluation method that directly estimates the model's confidence in predicting the correct tokens in a single forward pass. This approach offers a significant computational advantage, achieving up to 185x speedup compared to multi-sampling-based methods. Critically, this direct confidence score provides a more precise and fine-grained signal of the model's comprehension and potential for learning on a given sample, enabling a more accurate identification of the "fuzzy data" region. Leveraging this efficient uncertainty evaluation, we introduce the **UFO-RL** (**U**ncertainty-**F**ocused **O**ptimization for Efficient **R**einforcement **L**earning Data Selection) framework, designed to proactively select training samples from the model's dynamic ZPD, thereby significantly optimizing the efficiency and effectiveness of the RL fine-tuning process.

We conducted extensive empirical validation across multiple mathematical reasoning benchmarks and diverse LLM architectures of varying scales. Our results demonstrate the remarkable efficiency of UFO-RL: by judiciously selecting merely **10%** of the available data, UFO-RL requires less than **1/16** of the computational resources needed for full-data training, yet achieves performance comparable to, and often exceeding, the full-data baseline. Furthermore, UFO-RL exhibits enhanced training stability and improved generalization capabilities to unseen problems.

**The main contributions of this work are summarized as follows:**

- We introduce a novel principle for efficient RL fine-tuning data selection for LLMs, grounded in cognitive ZPD theory, advocating focus on data exhibiting intermediate model uncertainty.

- We propose **UFO-RL**, a lightweight and scalable framework that implements this principle, leveraging a novel, efficient single-pass uncertainty evaluation method to drastically reduce computational overhead compared to multi-sampling alternatives.

- We provide extensive empirical evidence, demonstrating state-of-the-art training efficiency across multiple benchmarks and LLM architectures: matching or surpassing full-data performance using only **10%** data and less than **1/16** computational resources, coupled with improved training stability and generalization.

## 2 Related Work

**RL-based Post-training for LLMs**  RL has become a key technique for fine-tuning LLMs, particularly on complex tasks such as mathematical reasoning [18, 4, 17]. Standard RL algorithms, including policy gradient methods like PPO [14], are computationally intensive primarily because they require multiple samples per instance to estimate gradients accurately and reduce variance. Methods like GRPO [16] improve training efficiency by leveraging group-relative policy optimization. Building on this, DAPO [23] further enhances efficiency by filtering out samples that are consistently answered correctly or incorrectly across multiple samplings. However, despite these improvements, DAPO still depends on computationally expensive multi-sampling to identify such samples, which limits scalability. In contrast, our approach tackles this fundamental bottleneck by introducing a novel single-pass uncertainty estimation method that efficiently evaluates data without requiring repeated sampling, thereby significantly reducing computational overhead during data selection.

**Data-Driven "Less is More"**  Beyond algorithmic optimization, data has emerged as a pivotal factor in acquiring and enhancing LLM capabilities [25, 24]. The "less is more" principle highlights that carefully selecting high-quality data can substantially improve training efficiency[19, 22, 26]. For instance, in the Supervised Fine-Tuning (SFT) stage, empirical evidence shows that using small, high-quality datasets [27, 22] can effectively enhance model performance, even surpassing fine-tuning on massive amounts of ordinary data, thereby achieving a significant leap in LLM capabilities. Notably, such high-quality data often correspond to complex samples, and prioritizing these challenging examples can boost reasoning abilities [20, 12].

However, identifying and leveraging "high-quality data" within the context of RL poses unique and significant challenges. While methods like LIMR [10] attempt data selection by focusing on samples with intermediate rewards or performance metrics—conceptually aligned with identifying challenging-but-achievable samples (similar to the Zone of Proximal Development, ZPD)—these approaches share a critical limitation: the computational cost of evaluating data remains prohibitively high. This is because estimating the value or informativeness of an instance in RL typically requires expensive multi-step interactions or rollouts to observe outcomes and compute rewards or performance signals. The resulting overhead makes it difficult to efficiently apply data selection strategies in large-scale RL processes and can offset or even undermine the efficiency gains brought about by data selection.

To overcome this challenge, our UFO-RL framework addresses this challenge with an efficient single-pass uncertainty estimation for data evaluation. This approach enables us to efficiently select informative samples (i.e., fuzzy data) in a scalable manner, completely avoiding the prohibitive computational cost of traditional methods.

## 3 Fuzzy data is beneficial for the learning process in RL.

This section presents a preliminary study that uses sampling accuracy as a quantifiable proxy for model uncertainty to investigate its relationship with RL learning outcomes, thereby motivating our uncertainty-aware approach.

### 3.1 Definition of Sampling Accuracy

Comparing with the ground-truth answer, a common and objective reward signal in RL for mathematical problems, allows us to gauge model consistency and infer the difficulty of individual data points by assessing answer accuracy across multiple generations.

Consider a dataset of $M$ examples, denoted as $\{x_1, \ldots, x_M\}$. To define the sampling accuracy $p_i$ for each example $x_i$, we generate $N = 16$ independent answers using a fixed LLM. Let $a_i^{(j)}$ be the $j$-th generated answer for $x_i$, and $y_i$ be the ground-truth answer. The sampling accuracy $p_i$ is then calculated as:

$$p_i = \frac{1}{N} \sum_{j=1}^{N} R\left(a_i^{(j)}, y_i\right),$$

where $R(\cdot)$ is the binary reward function (1 for correct, 0 otherwise). Intuitively, a lower $p_i$ indicates lower confidence in the model's responses for example $x_i$, suggesting that example $x_i$ is more difficult for the model. Conversely, a higher $p_i$ suggests higher confidence in the model's answers, suggesting that example $x_i$ is simpler for the model.

### 3.2 Experimental Setup

To empirically investigate the relationship between data difficulty and model learning efficiency, we conducted experiments using the GSM8K dataset [1] with several LLMs: four Qwen2.5 variants (0.5B, 1.5B, 3B, 7B) [21], Llama3.1-8B-Instruct [3], and Mistral-7B-Instruct [7]. The experimental procedure is as follows:

First, for each model, we computed the sampling accuracy $p_i$ for all examples in the GSM8K training set by performing $N = 16$ independent inference passes. During this process, the temperature was set to 1. $p_i$ thus serves as our preliminary proxy for model consistency and inferred example difficulty.

Next, to analyze how this proxy relates to learning, we sorted all training examples based on their computed $p_i$ values and partitioned them into $K = 10$ equally-sized bins $\mathcal{G}_0, \ldots, \mathcal{G}_9$. Each bin contains one-tenth of the total number of examples in the dataset. These bins are ordered by sampling accuracy from lowest to highest: Bin $\mathcal{G}_0$ contains the $10\%$ of examples with the lowest accuracy (corresponding to the highest inferred difficulty), Bin $\mathcal{G}_9$ contains the $10\%$ of examples with the highest accuracy (corresponding to the lowest inferred difficulty), while Bins $\mathcal{G}_1$ through $\mathcal{G}_8$ contain examples corresponding to the intermediate deciles.

Finally, using the Open-R1 RL fine-tuning framework [2], we conducted separate training runs **using only data sampled from each respective bin** ($\mathcal{G}_k$) to observe the learning efficiency achieved by training on data of different difficulty levels.

### 3.3 Results and Analysis

Figure 1 presents the learning effectiveness when training on data subsets corresponding to each sampling accuracy bin. The experiment demonstrates a clear non-monotonic relationship between inferred example difficulty (as proxied by sampling accuracy) and learning outcomes. Performance is generally lowest when training on the easiest data (highest accuracy bins), improves significantly when training on data of intermediate difficulty, and then declines again when training on the hardest data (lowest accuracy bins). This strongly supports the hypothesis that data points within the model's ZPD provide the most effective learning signal during RL fine-tuning. Intermediate sampling accuracy serves as a preliminary proxy for this optimal difficulty level.Analysis across models of varying scales further suggests that the optimal difficulty level for learning can be model-dependent, with larger models showing more robustness when trained on harder examples compared to smaller models.

Table 1: Distribution of Multiple Sampling Accuracy on the GSM8K Training Set for the Qwen2.5 family (0.5B, 1.5B, 3B, 7B), Llama3.1-8B-Instruct, and Mistral-7B-Instruct.

| Acc(%) | Qwen2.5-0.5B | Qwen2.5-1.5B | Qwen2.5-3B | Qwen2.5-7B | Llama3.1 | Mistral |
|---|---|---|---|---|---|---|
| 0 | 15.91 | 4.82 | 2.46 | 1.36 | 0.90 | 20.02 |
| [0, 15) | 37.31 | 15.83 | 8.26 | 3.21 | 2.40 | 34.20 |
| (85, 100] | 4.83 | 5.34 | 12.40 | 75.71 | 53.74 | 14.36 |
| 100 | 0.73 | 0.47 | 1.99 | 49.07 | 21.53 | 3.12 |

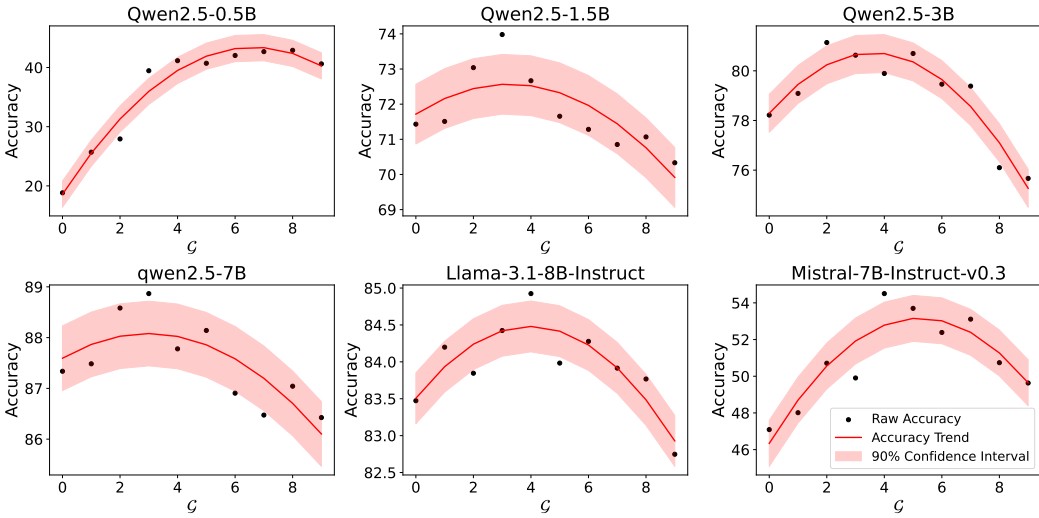

Figure 1: Impact of Sampling Accuracy-Based Data Subset Training on Model Learning Efficiency. Evaluation across Qwen2.5 (0.5B-7B), Llama3.1-8B, and Mistral 7B. GSM8K training data was divided into 10 bins by accuracy. Lines represent RL performance from exclusive training on each bin (lowest to highest accuracy), showing the relationship between inferred difficulty and learning efficiency.

A more detailed analysis of the distribution of multiple sampling accuracies for various models on the GSM8K dataset, as presented in Table 1, reveals important insights into the models' current capabilities on these problems. A considerable fraction of examples for many models is concentrated in the extreme accuracy bins (0% or 100%). For instance, a 0% accuracy rate (e.g., 20.02% for Mistral 7B) indicates instances that are consistently challenging or currently insurmountable for the model, reflecting a current failure mode. Conversely, a 100% accuracy rate (e.g., 49.07% for Qwen2.5-7B) suggests instances the model has mastered, indicating robust initial competence. This distribution of sampling accuracies, clearly correlating with the model's demonstrated ability to solve problems (from consistently failing to consistently succeeding), reinforces sampling accuracy as a suitable and intuitive proxy for data difficulty relative to the model's current state.

Understanding this difficulty profile is crucial. As suggested by the non-monotonic relationship observed in Figure 1, optimal learning is likely to occur on data of intermediate difficulty – those instances falling between these extremes. Consequently, focusing training efforts on data within this 'fuzzy' middle ground, where the model shows partial understanding but not mastery, is essential for efficient RL fine-tuning, underscoring the necessity of judicious data selection strategies.

# 4 Uncertainty-Focused Optimization for Efficient RL Data Selection

While sampling-based methods are effective in assessing data uncertainty, their prohibitive cost and coarse granularity limit their applicability in large-scale RL training. To address this, and to improve the efficiency and stability of RL for LLMs, we propose UFO-RL, a novel framework for strategic training data selection based on model uncertainty. This section first introduces our confidence estimation method, then validates its consistency with sampling-based metrics, analyzes its computational efficiency, and finally describes its application in uncertainty-driven data filtering.

## 4.1 Confidence Estimation via Average Log-Softmax

Although multi-sample accuracy provides a direct approach to estimating sample difficulty, it is computationally expensive and discrete, making it impractical for scaling RL training. To address this limitation, we propose an efficient confidence estimation technique based on token-level logit statistics obtained from a single forward pass.

Given an input $x_i$, we generate a complete answer sequence $\{y_1, y_2, \ldots, y_T\}$ using a decoder-only language model and record the conditional probability $P(y_t \mid x_i, y_{<t})$ at each decoding step. We define the confidence score for $x_i$ as the average log-probability across all output tokens:

$$\text{Conf}(x_i) = \frac{1}{T} \sum_{t=1}^{T} \log P(y_t \mid x_i, y_{<t}).$$

This confidence metric offers several practical advantages. First, it requires only a single forward pass, avoiding repeated inference. Second, it is fully parallelizable across batches and examples, making it highly scalable. Third, as demonstrated in section 4.2, it demonstrates a strong correlation with sampling accuracy. Furthermore, unlike reward functions, this approach provides a continuous uncertainty signal. This signal is lightweight and computationally inexpensive. Crucially, it is distinct from typical reward values; it is domain-agnostic yet model-specific, enabling its application across various reasoning tasks.

## 4.2 Comparison with Confidence and Sampling Accuracy

Table 2: Similarity Analysis of Multi-Sample Accuracy and Confidence Scores across the Qwen2.5 Family (0.5B, 1.5B, 3B, 7B), Llama-3.1-8B-Instruct, and Mistral-7B-Instruct-v0.3

| Model | Qwen2.5-0.5B | Qwen2.5-1.5B | Qwen2.5-3B | Qwen2.5-7B | Llama3.1 | Mistral |
|---|---|---|---|---|---|---|
| Similarity | 0.68 | 0.78 | 0.83 | 0.86 | 0.82 | 0.80 |

To validate that our confidence measure can serve as a useful and efficient proxy for estimating example difficulty, we conducted experiments to assess its consistency with sampling accuracy. As shown in Table 2, the confidence scores and multi-sample accuracy calculated for different models on the GSM8K training set exhibit high consistency, demonstrating strong alignment in ranking examples by inferred difficulty.

Table 3: Comparison of Qwen2.5-7B training performance using the easiest data deciles determined by multi-sample accuracy versus confidence.

| Method | 10% Easiest | 10%-20% Easiest | 20%-30% Easiest | 30%-40% Easiest |
|---|---|---|---|---|
| **Accuracy** | 90.37 | 89.92 | 90.51 | 89.87 |
| **Confidence** | 71.09 | 88.01 | 91.28 | 91.50 |

Furthermore, the discrete nature of accuracy and rule-based rewards limits their granularity for nuanced data selection. As illustrated in Table 1, a substantial portion of data for models like Qwen2.5-7B (49%) yields 100% accuracy even with multiple samples, making finer distinction based on accuracy impossible. Table 3 compares the learning performance when training on successive deciles of the 'easiest' data (lowest inferred difficulty), as identified separately by sampling accuracy and confidence. Accuracy-based selection shows minimal performance differentiation across these bins, whereas the confidence-based method reveals clear performance variations, indicating its superior ability to segment data difficulty within seemingly 'easy' ranges.

Computational efficiency is paramount for scaling RL to large models and datasets. Table 4 presents the computational cost comparison for evaluating difficulty across models using single-pass confidence vs. multi-sample accuracy (16 samples per instance, accelerated with VLLM [8]) on the GSM8K dataset. As shown, confidence estimation achieves a speedup of up to $185\times$ compared to multi-sample accuracy under equivalent resource conditions. This dramatic difference highlights its advantage as a practical evaluation metric, making confidence-based evaluation highly suitable for large-scale data filtering and dynamic data selection within RL training loops.

## 4.3 Confidence-Based Data Filtering

Drawing upon ZPD, we posit that the most informative samples reside within an intermediate difficulty spectrum that is contingent upon the model's current capabilities. To delineate this ZPD, a

Table 4: Computational Efficiency Comparison between Multi-Sample Accuracy and Confidence Estimation on a Single A100 GPU for Qwen2.5 family, Llama-3.1-8B-Instruct, and Mistral-7B-Instruct-v0.3. Our method achieves up to $185\times$ speedup over accuracy-based methods.

| Method | Model | Time | SpeedUp | Model | Time | SpeedUp |
|---|---|---|---|---|---|---|
| Accuracy
Confidence | Qwen2.5-0.5B | 3337s
18s | $\times 185$ | Qwen2.5-1.5B | 3712s
45s | $\times 82$ |
| Accuracy
Confidence | Qwen2.5-3B | 4186s
89s | $\times 47$ | Qwen2.5-7B | 6827s
175s | $\times 39$ |
| Accuracy
Confidence | Llama3.1-8B | 11426s
186s | $\times 61$ | Mistral 7B | 8335s
146s | $\times 57$ |

metric capable of fine-grained differentiation is necessary. To this end, we leverage the continuous nature of the confidence score $\text{Conf}(x_i)$ and define a derived score:

$$s_i = \exp(\text{Conf}(x_i))$$

, where $s_i \in (0, 1)$. This score can be interpreted as the geometric mean of the token probabilities, serving as a continuous measure of the model's certainty regarding the sample. Higher values indicate greater confidence, while lower values signify increased difficulty.

For each sample in the dataset, we compute a "fuzziness score" to evaluate its suitability for training. This score is formally defined as: $\text{Score}(s_i) = 1 - (s_i - \mu)^2$, where $\mu$ represents the mean confidence score of the candidate dataset. This function is designed to assign higher scores to samples whose confidence $s_i$ is close to the mean confidence $\mu$, reflecting the moderate uncertainty characteristic of samples within the ZPD.

Ultimately, the top 10% of samples, as ranked by this "fuzziness score", are selected from the candidate dataset to constitute the training data for RL.

## 5 Experiments and Analyses

To evaluate the effectiveness and efficiency of uncertainty-driven sample selection in RL, we conducted a comprehensive empirical study across multiple language models and benchmark datasets. This section provides a detailed description of the experimental setup, training details, core performance findings, and computational cost analysis.

### 5.1 Training Setup

Our experiments utilized several models ranging from 0.5B to 8B parameters. Data selection was conducted on the commonly used RL datasets GSM8K [1] and DAPO-MATH-17K [23]. Training was performed using the open-source GRPO framework from open-r1. Details regarding models, datasets, training environment, and hyperparameters are provided in Appendices A.3, A.4, and A.5.

For comparison, we conducted experiments evaluating several distinct data selection strategies for RL fine-tuning. These strategies aimed to assess the impact of different data subsets on model performance and training efficiency. We included the **Full Data** strategy, training using the entire available training dataset as the standard performance and efficiency baseline. We also evaluated training on a 10% subset of examples exhibiting the highest confidence scores, termed **High Conf**, representing focus on data the model already finds 'easy'. Conversely, we assessed training on a 10% subset with the lowest confidence scores, termed **Low Conf**, representing focus on data the model finds 'hard' or is highly uncertain about. As a baseline for data reduction without targeted selection, we used **Random** sampling, training on a 10% uniformly random subset. Furthermore, we included **Acc$_{\text{Filter}}$**, training on data remaining after filtering out examples with extreme multi-sample accuracy (precisely 0% or 100%) to explore excluding consistently failed or mastered data. Finally, we evaluated **UFO$_{\text{Ours}}$**, our proposed method, which trains on a 10% subset specifically selected based on intermediate uncertainty using our efficient confidence approach, consistent with the Zone of Proximal Development (ZPD) principle. To ensure robustness, the results for the **Random** sampling strategy were averaged over 5 independent runs.

Table 5: Performance of Different Data Selection Strategies. Performance is measured by accuracy (%). Results are reported separately for models trained on the GSM8K and DAPO-MATH-17K datasets. We evaluated several strategies: **Full Data** (baseline), **High Conf**, **Low Conf**, **Random**, **Acc$_{Filter}$**, and **UFO$_{Ours}$**.

| Model | Dataset | Full Data | High Conf | Low Conf | Random | Acc$_{Filter}$ | UFO$_{Ours}$ |
|---|---|---|---|---|---|---|---|
| *Trained on GSM8K* | | | | | | | |
| Qwen2.5-0.5B | GSM8K | 52.66 | 45.67 | 41.57 | 41.96 | **50.61** | 46.27 |
| | Math500 | 30.80 | 30.40 | 28.20 | 29.64 | 30.40 | **30.60** |
| | MMLU | 42.06 | 41.90 | 41.93 | 41.97 | 41.94 | **42.14** |
| Qwen2.5-1.5B | GSM8K | 76.78 | 73.22 | 74.12 | 74.83 | 76.48 | **76.63** |
| | Math500 | 53.00 | 54.00 | 54.00 | 54.40 | 55.20 | **55.40** |
| | MMLU | 57.86 | 58.06 | 58.20 | 58.14 | 58.01 | **58.32** |
| Qwen2.5-3B | GSM8K | 83.33 | 62.67 | 82.17 | 82.25 | 83.61 | **84.37** |
| | Math500 | 66.20 | 65.00 | 64.40 | 65.84 | 65.40 | **67.40** |
| | MMLU | 64.56 | 64.51 | 64.61 | 64.55 | **64.89** | 64.65 |
| Qwen2.5-7B | GSM8K | 91.88 | 71.09 | 91.20 | 90.93 | 91.35 | **92.03** |
| | Math500 | 75.00 | 74.40 | 75.60 | 75.64 | 76.20 | **76.40** |
| | MMLU | 69.64 | 69.25 | **69.44** | 69.14 | 69.26 | 69.43 |
| LLaMA3.1-8B | GSM8K | 88.62 | 86.05 | 87.55 | 87.29 | 88.01 | **88.30** |
| | Math500 | 51.00 | 50.80 | 52.00 | 53.14 | 52.00 | **55.00** |
| | MMLU | 62.63 | 59.34 | 61.65 | 60.05 | 59.12 | **61.67** |
| Mistral-7B | GSM8K | 53.87 | 51.74 | 47.95 | 52.81 | 55.46 | **56.67** |
| | Math500 | 11.60 | 13.40 | 12.00 | 13.14 | 13.80 | **14.20** |
| | MMLU | 58.46 | 59.38 | 59.44 | 59.61 | 58.96 | **59.66** |
| Model | Dataset | Full Data | High Conf | Low Conf | Random | Acc$_{Filter}$ | UFO$_{Ours}$ |
| *Trained on DAPO-MATH-17K* | | | | | | | |
| Qwen2.5-0.5B | GSM8K | 12.44 | 40.82 | 17.06 | 35.71 | 14.26 | **41.43** |
| | Math500 | 29.40 | 30.02 | 28.60 | 29.80 | 29.20 | **33.40** |
| | MMLU | 42.28 | 42.17 | 42.17 | 42.19 | **42.27** | 42.18 |
| Qwen2.5-1.5B | GSM8K | 55.01 | 55.84 | 47.95 | 49.27 | 55.39 | **55.87** |
| | Math500 | 54.20 | 53.60 | 52.00 | 52.94 | 54.60 | **54.80** |
| | MMLU | 58.28 | 58.30 | 58.11 | 58.22 | **58.46** | 58.17 |
| Qwen2.5-3B | GSM8K | 60.69 | 60.55 | 50.70 | 61.81 | 60.77 | **65.70** |
| | Math500 | 63.60 | 66.00 | 65.80 | 66.24 | 67.60 | **68.20** |
| | MMLU | 64.83 | **64.63** | 64.57 | 64.62 | 64.57 | 64.55 |
| Qwen2.5-7B | GSM8K | 92.03 | 84.75 | 81.34 | 87.43 | 88.09 | **91.16** |
| | Math500 | 75.80 | 76.40 | 77.20 | 75.46 | 75.60 | **77.40** |
| | MMLU | 70.33 | 68.88 | 68.88 | 69.17 | 68.91 | **69.69** |
| LLaMA3.1-8B | GSM8K | 86.56 | 59.41 | 86.12 | 82.67 | 86.12 | **86.20** |
| | Math500 | 45.80 | 47.40 | 49.80 | 49.40 | 51.80 | **52.40** |
| | MMLU | 61.47 | 60.30 | 60.64 | **63.06** | 60.78 | 59.59 |
| Mistral-7B | GSM8K | 44.40 | 49.54 | 48.72 | 47.60 | 49.62 | **50.38** |
| | Math500 | 10.60 | 12.40 | 11.80 | 12.38 | 11.60 | **14.80** |
| | MMLU | 58.71 | **59.78** | 58.95 | 59.28 | 59.30 | 59.11 |

## 5.2 Main Results: Performance Evaluation

In Table 5, we conducted two main sets of experiments based on the training dataset. When trained on GSM8K, our **UFO** strategy demonstrates robust performance across various task domains:

- On the **in-domain** GSM8K test set, UFO consistently achieves results comparable to or slightly exceeding full-data fine-tuning across all evaluated model scales.
- On the **near-domain** Math500 benchmark, UFO generally outperforms full-data RL, indicating enhanced generalization to more challenging mathematical reasoning.

- On the **out-of-domain** MMLU benchmark, performance variations across different data selection strategies are relatively marginal, suggesting a limited impact of domain-specific RL on general language understanding, irrespective of the selection mechanism.

Analysis of the baseline strategies reveals patterns consistent with our central hypothesis. On the one hand, training exclusively on high-confidence samples, compared to training on a random subset, frequently leads to performance stagnation or even degradation, particularly pronounced for larger models, likely due to overfitting on overly simplistic instances that offer diminished learning signals. On the other hand, training on low-confidence samples tends to destabilize the optimization process, especially evident in models with lower capacity. In contrast to these detrimental extremes, strategies like random sampling and $\text{Acc}_{\textbf{Filter}}$ show improved performance. Specifically, the $\text{Acc}_{\textbf{Filter}}$ method (which trains on data after removing examples with 0% or 100% multi-sample accuracy) demonstrates that excluding these extreme examples is an effective approach to improve RL efficiency. However, while random sampling and $\text{Acc}_{\textbf{Filter}}$ strategies generally outperform training on extreme confidence subsets, they mostly underperform our UFO approach, which specifically targets the intermediate uncertainty range. Therefore, this underscores the efficacy of targeted data selection over indiscriminate sampling.

To further substantiate the generalizability of our proposed methodology, we replicated our experimental protocol on the more challenging and diverse DAPO-MATH-17K dataset, which presents a higher degree of complexity compared to the elementary-level mathematics of GSM8K. The results obtained on DAPO-MATH-17K mirrored the trends observed on GSM8K, providing additional empirical validation for our central thesis. Notably, on the challenging DAPO-MATH-17K dataset, which can potentially cause catastrophic effects for smaller models (as mentioned in [9]), our method demonstrates resilience and avoids such drastic performance drops seen with less targeted selection or on smaller data subsets in some cases, further highlighting the benefits of focusing training on informative data.

### 5.3 Computational Cost Analysis

Beyond performance benefits, a key advantage of UFO-RL lies in its efficiency. Prior work [13, 4] has demonstrated that the computational cost per instance tends to increase significantly as RL progresses, primarily because the inference length, especially with multi-sampling, grows considerably during training. Our method leverages effective data selection to achieve significant performance with only 10% of the data, which leads to a drastic reduction in the overall computational cost of RL fine-tuning. As detailed in Table 6, our method achieves a speedup of up to $16\times$ compared to RL on the full dataset. This substantial speedup is a direct result of processing significantly fewer training instances in the RL loop, thereby mitigating the amplified cost associated with the longer inference sequences in later training stages.

Table 6: RL Fine-tuning Time Comparison: UFO (10% Mid Confidence Data) vs. Full Data (in seconds, on an 8 x A100 GPU server).

| Method | Model | Time (s) | Speedup | Method | Time (s) | Speedup |
|---|---|---|---|---|---|---|
| UFO
Full Data | Qwen2.5-0.5B | 140
1815 | $\times 13$ | Qwen2.5-1.5B | 407
5694 | $\times 14$ |
| UFO
Full Data | Qwen2.5-3B | 739
10224 | $\times 14$ | Qwen2.5-7B | 1154
12959 | $\times 11$ |
| UFO
Full Data | Llama3.1-8B | 1219
14040 | $\times 12$ | Mistral 7B | 1454
22955 | $\times 16$ |

## 6 Conclusion

In conclusion, this work introduces UFO-RL, a novel and efficient framework for RL fine-tuning of LLMs, inspired by the ZPD theory. By employing a computationally lightweight single-pass uncertainty estimation method, UFO-RL efficiently identifies and prioritizes training samples within the model's "fuzzy data" regime, where it exhibits the most significant potential for learning. Our comprehensive empirical evaluation across diverse mathematical reasoning benchmarks and various LLM architectures demonstrates the remarkable efficiency and effectiveness of UFO-RL. Specifically,

we demonstrate that UFO-RL achieves performance comparable to, and often exceeding, full-data training by training on only 10% of the data. This targeted selection results in a drastic reduction in computational cost, requiring less than 1/16 of the resources needed for full-data training. Furthermore, our findings indicate that UFO-RL enhances training stability and improves generalization to unseen instances. These results strongly suggest that focusing RL fine-tuning on samples within the model's estimated ZPD offers a promising avenue for significantly improving the efficiency and efficacy of training LLMs for complex reasoning tasks.

## 7 Acknowledgements

The research in this article is supported by the New Generation Artificial Intelligence of China (2024YFE0203700), National Natural Science Foundation of China under Grants U22B2059 and 62176079.

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

# A   Technical Appendices and Supplementary Material

## A.1   Limitations

Although the UFO-RL method proposed in this paper has been effectively validated across multiple models and datasets, due to the substantial computational overhead of RL, our experiments were limited to smaller-scale language models (up to 8B parameters) and relatively small training datasets (up to 17K examples). The computational and training costs of RL significantly increase when scaling up to larger models (e.g., those with more than 8B parameters) and larger datasets (e.g., those containing millions or tens of millions of samples). As a result, experiments on these larger-scale models and datasets were not conducted in this work. Therefore, the experimental results presented in this paper do not cover the scenario of large-scale models and datasets, which remains an open challenge.

## A.2   Broader Impacts

The UFO-RL method significantly reduces the computational resources required for RL-based fine-tuning of LLMs by improving data selection efficiency and reducing costly multi-sampling. This leads to faster training times and a substantial reduction in energy consumption, aligning with the principles of Green AI and promoting more sustainable AI development. It lowers financial and environmental costs, enhancing the accessibility of cutting-edge AI research for institutions with limited resources.

Potential Negative Impacts: While UFO-RL offers significant efficiency advantages, we also considered potential negative societal impacts. A primary risk is that increasing the efficiency of LLM fine-tuning could lower the cost and technical barrier for malicious actors to fine-tune existing base models for harmful purposes (e.g., generating misinformation, toxic content), if these base models themselves pose such risks. Nevertheless, our work primarily focuses on data selection in the domain of mathematical reasoning, where the direct negative societal impact is typically less severe than in general text generation. Furthermore, our method focuses on optimizing the learning process and does not inherently introduce new harmful capabilities.

## A.3   Model Details

For our experiments, we selected a range of prevalent language models to evaluate. Specifically, we utilized models from the Qwen2.5 family (0.5B, 1.5B, 3B, 7B) [21]. The Qwen series is commonly employed in various RL training contexts. [2, 13] Additionally, we included Llama3.1-8B-instruct, a widely recognized model known for its strong capabilities across many tasks, and Mistral-7B-Instructv0.3, a more established architecture which has shown relatively average performance on some mathematical benchmarks. The selection covers models with varying parameter counts and architectural characteristics to assess performance broadly.

## A.4   Dataset Details

Our experiments utilized specific datasets for both training and evaluation. For training, we primarily used the training split of GSM8K [1], a widely adopted dataset for enhancing models' mathematical abilities through RL, which comprises relatively simple elementary school-level math problems. We complemented this with DAPO-MATH-17K[23], a more recent dataset focused on RL for math, featuring a broader range of problem types and generally considered more difficult.

For evaluation, we assessed models on the GSM8K test set, which contains elementary math problems. To gauge performance on more complex mathematical and quantitative reasoning tasks, we included Math500[11]. Furthermore, MMLU [5]was used as a general-domain text understanding benchmark to evaluate the models' overall capabilities and generalization ability beyond mathematical tasks.

## A.5   Training and Evaluation Details

During the training phase, all experiments were conducted using the open-r1 framework, executed on a computing cluster equipped with 8 NVIDIA A100 GPUs.Key parameters included a learning rate of 1e-6.Training acceleration was achieved through the use of DeepSpeed Zero-2 optimization

technology. During the training process, seven responses were generated for each problem, with response generation accelerated using vLLM.

In the model evaluation phase, all our experiments employed a zero-shot evaluation method. To ensure fairness, the temperature was set to 0. Additionally, we utilized vLLM to accelerate inference, thereby improving efficiency.

