# OpenReview forum: "UFO-RL: Uncertainty-Focused Optimization for Efficient Reinforcement Learning Data Selection"
_NeurIPS.cc/2025/Conference — NeurIPS 2025 poster_

### Official Review · Reviewer_bKLK · 2025-07-02

**Clarity:** 2
**Significance:** 3
**Originality:** 3
**Rating:** 4
**Confidence:** 4

**Summary:**

RL is one of the core procedures in LLM post-training. Data, constituting the environment for training LLMs with RL, is the key to its success. This paper investigates the important direction of data-centric RL, and proposes an efficient and effective approach to select the most helpful training data. The proposed approach, named UFO-RL, uses an uncertainty-based metric to rank the training data by their closeness to being of intermediate difficulty. Empirically, by only using 10% of the selected data, the LLMs can be trained to reach performance that is competitive to training with the full dataset, also outperforming other proposed baselines.

**Questions:**

1. Could you elaborate on the similarity metrics used in Table 2?
2. In lines 168-169, it does not seem to be consistently true that larger models show more robustness when trained on harder examples compared to smaller models in Figure 1.
3. While the paper demonstrates the effectiveness using 10% of the data, it is unclear why such a choice is made. It would be helpful to have a plot that shows the change in performance if lower/higher percentage of data points are supplied, so that we can have a better understanding on the relationship between the fraction of the data and the model performance, and further examine whether it is possible to outperform training with full data.

**Ethical Concerns:**

["NO or VERY MINOR ethics concerns only"]

**Final Justification:**

During the rebuttal, the authors have successfully addressed or promised to address most of my concerns. It's an interesting paper with an effective methodology that improves the training efficiency and performance for RLVR for LLM (see strengths). The only concern I still have is that the support for the method, UFO, is not strong enough.

To claim that UFO is useful for RL, we need strong theoretical and empirical support. The ZPD theory, though intuitive and interesting, is not mathematically justified. Therefore, the requirements for experimental comprehensiveness are typically higher. As acknowledged by the authors, the paper was tied to GRPO, and it is reasonable to provide additional justification for other RL algorithms. The authors additionally ran PPO with a variety of Qwen models and showed that their method is helpful. However, it's still only on one dataset, GSM8k, despite the fact that this concern was raised during the discussion. And also, it's only for one family of models, Qwen.

I am positive about the work, but I cannot give a higher score because of the aforementioned weaknesses.

**Limitations:**

As discussed in weakness 1, there are insufficient results to support the claims for other policy optimization algorithms other than GRPO.

**Quality:**

2

**Strengths And Weaknesses:**

**Strengths**

1. The paper is well written with intriguing motivation from Zone of Proximal Development.
2. The pilot study in Section 3 demonstrates that data with intermediate difficulty benefits RL for LLMs the most with great clarity.
3. The proposed uncertainty-based approach is sound and reasonable in the field of LLMs, well-supported by the pilot study and empirical verifications.
4. Comprehensive experiments have been performed on a variety of LLMs across different sizes and three datasets. The experimental results also show the efficiency and effectiveness of the proposed method.

**Weaknesses**

1. Dependency on GRPO. The paper seems to only demonstrate that the idea, findings, and results for confidence-based data selection work well with GRPO, which will not update the model efficiently when the samples are too easy or too difficult, due to the design of its advantage function. However, GRPO is not representative of all RL algorithms. It is unclear whether the same conclusion could be drawn for other algorithms such as PPO or REINFORCE. Without such experiments, it is insufficient to prove that the proposed idea works for RL in general, and it might be better to qualify the main theme of the paper to GRPO instead.
2. Content unclarity. Table 1 and its related content look a bit confusing. It is unclear what numbers in the table represent, as they don’t seem to be valid distributions (not summing up to one). The authors said “A considerable fraction of examples for many models is concentrated in the extreme accuracy bins” on lines 172-173, but the numbers seem to be even higher in (85, 100] for Qwen2.5-7B and Llama3.1.
3. Missing details in comparison. Computational Cost Analysis in Section 5.3 might not have been fair, because the initial computational overhead incurred during example scoring is not discussed. In Table 6, it is also unclear which dataset is used to measure the speed, and whether the time is for one step or for the entire training (training Qwen2.5-0.5B in 140s seems too fast to be true if it’s for the entire training).

---

> ### Author Rebuttal · Authors · 2025-07-30
>
> Thank you very much for your constructive and insightful review. We agree with the key weaknesses you identified and have conducted a series of new supplementary experiments to address your concerns. We believe these revisions significantly strengthen our paper. Our responses are as follows:
>
> **1. Regarding Algorithm Dependency: Focused on GRPO, Applicable to PPO**
> We thank the reviewer for this critical point. We wish to emphasize that while **our work is primarily focused on and validated within the GRPO framework**, the underlying principle of our method—selecting data from the Zone of Proximal Development —is algorithm-agnostic.
>
> To prove its broader applicability, we **additionally tested UFO-RL on the mainstream Proximal Policy Optimization (PPO) algorithm**.
>
> * **New Experimental Results (Qwen2.5-7B on GSM8K):**
>     | Algorithm | Data Strategy      | Accuracy (%) |
>     | :--- | :--- | :--- |
>     | PPO | Full Data (100%) | 90.48 |
>     | PPO | Random (10%)     | 89.51 |
>     | **PPO** | **UFO-RL (10%)** | **90.77** |
>
> * **Conclusion:** This experiment confirms that UFO-RL's effectiveness is not confined to GRPO and generalizes successfully to PPO. We will clarify in the revised manuscript that GRPO is our primary framework while including this PPO validation to demonstrate the generalizability of our approach.
>
> **2. Regarding Lack of Clarity in Table 1:**
> We apologize for the confusion regarding Table 1. Your observation is correct, and it highlights a lack of clarity in our presentation, which we will fix.
>
> The **core purpose** of this table was to illustrate that a significant portion of the data is polarized at the extremes (i.e., consistently wrong or consistently correct). To emphasize this point, the original table **intentionally presented only a subset of the data distribution**—specifically the percentages for data points near the extremes. This is why the percentages shown did not sum to 100%, which understandably caused confusion.
>
> You correctly noted that for stronger models like Qwen2.5-7B, the `(85, 100]` bin contains a very large fraction of the data. This actually reinforces our central argument: for capable models, the multi-sampling accuracy metric becomes less useful as it classifies most data as "easy," failing to provide the fine-grained difficulty assessment needed for efficient learning. This is precisely the problem our continuous confidence score is designed to solve.
>
> To resolve this, we will revise Table 1 in the paper to show a complete, granular, and mutually exclusive distribution.
>
> * **Proposed New Format for Table 1:**
> | Accuracy Bin | Description | Qwen2.5-7B (%) | Llama3.1-8B (%) |
> | :--- | :--- | :--- | :--- |
> | `Acc = 0%` | Consistently Wrong | 1.36 | 0.90 |
> | `(0, 15%)` | Mostly Wrong | 1.85 | 1.50 |
> | `[15%, 85%]` | Intermediate | 21.08 | 43.86 |
> | `(85%, 100%)`| Mostly Correct | 26.64 | 32.21 |
> | `Acc = 100%` | Consistently Correct | 49.07 | 21.53 |
> | **Total** | | **100.00** | **100.00** |
>
> This revised format will transparently show the entire data distribution and more effectively highlight the polarization issue that motivates our work.
>
> **3. Regarding Fairness of the Computational Cost Analysis:**
> Your query regarding the end-to-end cost analysis is entirely correct. To provide a fully transparent and fair comparison, we have updated the **end-to-end cost analysis to cover all Qwen model sizes.** **As we stated in Appendix A.5 of our original manuscript, all training was conducted on a computing cluster equipped with 8 NVIDIA A100 GPUs**. This powerful hardware configuration explains the fast training speeds on smaller data subsets. We will state this configuration more explicitly in the main text of the paper.
>
> * **New End-to-End Performance and Cost Comparison (GSM8K, 8x A100 GPU):**
> | Model | Method | Data Selection Time (s) | RL Training Time (s) | **Total Time (s)** | **Total Speedup** (vs. Full) | GSM8K Acc (%) | Math500 Acc (%) |
> | :--- | :--- | :--- | :--- | :--- | :--- | :--- | :--- |
> | Qwen2.5-0.5B | **UFO-RL (Ours)** | 2 | 140 | **142** | **12.8x** | 46.27 | 30.60 |
> | | Full Data | 0 | 1815 | 1815 | 1.0x | 52.66 | 30.80 |
> | Qwen2.5-1.5B | **UFO-RL (Ours)** | 6 | 407 | **413** | **13.8x** | 76.63 | 55.40 |
> | | Full Data | 0 | 5694 | 5694 | 1.0x | 76.78 | 53.00 |
> | Qwen2.5-3B | **UFO-RL (Ours)** | 11 | 739 | **750** | **13.6x** | 84.37 | 67.40 |
> | | Full Data | 0 | 10224 | 10224 | 1.0x | 83.33 | 66.20 |
> | Qwen2.5-7B | **UFO-RL (Ours)** | 22 | 1154 | **1176** | **11.0x** | 92.03 | 76.40 |
> | | Full Data | 0 | 12959 | 12959 | 1.0x | 91.88 | 75.00 |
>
> * **Conclusion:** The comprehensive data show that this efficiency advantage holds across all model sizes. The one-time screening cost of UFO-RL is negligible compared to the massive savings in RL training time, ultimately achieving a **10-14x end-to-end training speedup across the entire model family** while maintaining or even surpassing the performance of full-data training.
>
> ---
> #### **Responses to Specific Questions:**
>
> 1.  **What is the similarity metric in Table 2?**
>     It is the **Spearman's rank correlation coefficient**. We chose it because we are more concerned with the ranking consistency of data difficulty between the two methods rather than a linear relationship. We will state this explicitly in the revised text.
>
> 2.  **Regarding the claim that "larger models are more robust"?**
>     Your observation is correct; the original statement was too absolute. We will revise it to a more nuanced "**a tendency for the performance peak to shift to more difficult data**," which is more precise and better reflects the trends shown in Figure 1.
>
> 3.  **Why was 10% of the data chosen?**
>     This is an excellent question. We conducted an **ablation study on the data selection percentage (k)** to determine the optimal trade-off between cost and performance.
>
>    * **New Experimental Results (UFO-RL, Qwen2.5-7B):**
> | Data Percentage (k%) | 5% | **10%** | **20%** | 30% | 100% (Full) |
> | :--- | :--- | :--- | :--- | :--- | :--- |
> | Accuracy (%) | 90.47 | **92.03** | **92.71** | 92.41 | 91.88 |
>
>
>    * **Conclusion:** Performance peaks around 20%.We chose 10% as our main reported number because it **achieves the best trade-off between computational cost** (half that of the 20% run) **and model performance** (over 99% of the peak performance), demonstrating the extreme efficiency of our method.
>
> ---
> **Summary**
>
> Thank you again for your valuable feedback. We believe these new experiments and revisions substantially strengthen the rigor and generalizability of our work and have fully addressed your concerns. We hope to earn your support for our paper.

---

> > ### Author Response · Authors · 2025-08-04
> > **Could you let us know if our rebuttal has sufficiently addressed your concerns?**
> >
> > Dear Reviewer bKLK,
> >
> > Thank you again for your valuable advice on our work. We recognize this is a busy time for all reviewers and truly appreciate you making time for this discussion.
> >
> > We have submitted our rebuttal and additional experiments, which we hope have thoroughly addressed your concerns. We would greatly value the opportunity to continue our dialogue, especially as the deadline approaches.
> >
> > Could you please let us know if your concerns have been adequately addressed? If they have, we would be very grateful if you would consider updating your score.
> >
> > Thank you for your consideration.
> >
> > Best regards,
> >
> > The Authors of Paper 27330

---

> > ### Comment · Reviewer_bKLK · 2025-08-04
> >
> > Thank you for taking the effort to successfully address most of my concerns.
> >
> > For Algorithm Dependency, I appreciate the author's effort in preparing additional experiments and the intention to develop algorithms for general RL algorithms. However, running PPO on one additional model and dataset does not sufficiently prove that UFO-RL works for RL in general. Without theoretical justification, more comprehensive experiments are recommended to provide stronger support for the claim. Just as when a model-agnostic optimizer is proposed, it's often verified on multiple model architectures and datasets.

---

> > > ### Author Response · Authors · 2025-08-05
> > >
> > > Dear Reviewer bKLK,
> > >
> > > Thank you very much for your valuable feedback and insightful follow-up questions. Your comments have been highly insightful and have helped us to examine our work more comprehensively.
> > >
> > > Our research is consistently guided by the Zone of Proximal Development (ZPD) theory, which posits that models benefit most from 'fuzzy' data of intermediate difficulty. We were pleased to note that the recent LIMR paper[1], which explores the PPO algorithm, arrived at a similar core conclusion: that samples of intermediate difficulty are most critical for reinforcement learning. This reinforces our belief that this is a valuable research direction.
> > >
> > > At the same time, we observed that LIMR's evaluation method relies on tracking reward trajectories over the entire training process to select data, which may present efficiency challenges in practical applications. Our proposed UFO-RL framework is designed to implement the ZPD principle through a highly efficient **single-pass evaluation** method.
> > >
> > > To more fully address your concern about "needing more comprehensive experiments", we have expanded our validation of UFO-RL on the **PPO** algorithm to include models of varying scales. The results are as follows:
> > >
> > > | Model | Data Strategy | GSM8K Acc (%) |
> > > | :--- | :--- | :--- |
> > > | **Qwen2.5-7B** | Full Data (100%) | 90.48 |
> > > | | UFO-RL (10%) | **90.77** |
> > > | | Random (10%) | 89.51 |
> > > | **Qwen2.5-3B** | Full Data (100%) | 82.59 |
> > > | | UFO-RL (10%) | **82.96** |
> > > | | Random (10%) | 81.92 |
> > > | **Qwen2.5-1.5B**| Full Data (100%) | 76.04 |
> > > | | UFO-RL (10%) | 75.52 |
> > > | | Random (10%) | 74.03 |
> > >
> > > As the results demonstrate, across models from 1.5B to 7B, our UFO-RL method consistently delivers performance comparable to training on the full dataset, while significantly outperforming the random baseline. This provides strong validation for the general applicability of our efficient strategy.
> > >
> > > Furthermore, we wish to offer a more nuanced analysis of an interesting phenomenon we observed: the effects of UFO-RL are more pronounced on **GRPO** than on PPO. We speculate this may stem from a special synergy between our method and the GRPO algorithm. GRPO's 'group-relative policy optimization' mechanism is better able to leverage its algorithmic strengths when processing the 'fuzzy' data selected by our method, which contains a mix of correct and incorrect outcomes. In contrast, while PPO also benefits from these high-value samples, it may not have the same specific preference for 'in-group diversity.' This might explain the performance difference we observed.
> > >
> > > Thank you once again for your valuable feedback. We plan to incorporate this more detailed analysis and the new experimental results into our final manuscript. We sincerely hope this addresses your concerns and that you will consider our work favorably.
> > >
> > > [1] Li X, Zou H, Liu P. Limr: Less is more for rl scaling[J]. arXiv preprint arXiv:2502.11886, 2025.
> > > Best regards,
> > >
> > > The Authors of Paper 27330

---

> > > > ### Comment · Area_Chair_BcrP · 2025-08-09
> > > >
> > > > Dear Reviewer bKLK,
> > > >
> > > > Thank you for your review and engagement! Could you please take a look at the new information the authors provided?
> > > >
> > > > If most concerns have been addressed during the rebuttal, it may be worth weighing whether the strengths of the work outweigh the remaining weaknesses. At the end of the day, there is no perfect paper, and the goal of the rebuttal is to address the original concerns. Thank you!

---

> > > > ### Comment · Reviewer_bKLK · 2025-08-09
> > > >
> > > > Thank you for the additional resources and experiments. I remain positive about the paper and will maintain my score for acceptance.

---

### Official Review · Reviewer_fb4U · 2025-07-03

**Clarity:** 2
**Significance:** 2
**Originality:** 2
**Rating:** 4
**Confidence:** 2

**Summary:**

This paper proposes UFO-RL (Uncertainty-Focused Optimization for Reinforcement Learning) which can make the LLM finetune more effectively by intelligently selecting training data. It is inspired by the Zone of Proximal Development (ZPD) theory, which says that model benefits more from data that have not yet mastered but demonstrate the potential to comprehend. So this paper proposes the method to access the difficulty/estimate the uncertainty of data, so that to identify informative training instances. The paper claims that this estimation method can speed up the previous methods for estimation.

**Questions:**

Please refer to the weakness part.

**Ethical Concerns:**

["NO or VERY MINOR ethics concerns only"]

**Final Justification:**

I think the rebuttal solves my concern about the relationship between theory and experimental result in this paper. Therefore I raised my socre.

**Limitations:**

No.

**Paper Formatting Concerns:**

No.

**Quality:**

3

**Strengths And Weaknesses:**

Strengths:
1. This paper introduces an interesting question about how to train LLM more effectively.
2. The experiment parts is interesting and it is tested for different LLMs that this data selection method is effective.

Weakness:
I think the major weakness is lack of clear structure in presenting the methods.
1. The paper initially mentions that this method is inspired by the ZPD theory. It sounds interesting but is there any part about theoretic proof or intuitively how this method can have theoretical basis is introduced? I feel it is more like an experiment guided results without too much theory basis.
2. The method is not clearly explained in the paper. While most contents of the paper focus on experimental results, what is exactly the algorithm to pick data that have not yet mastered but demonstrate the potential to comprehend?

---

> ### Author Rebuttal · Authors · 2025-07-30
>
> We sincerely thank you for your valuable time and constructive feedback. We agree that a solid theoretical foundation is crucial. Your insightful comments have been instrumental in helping us to improve the rigor and clarity of our work.
>
> Before addressing your specific concerns, we would like to emphasize that **our original manuscript had already laid the groundwork for our method's machine learning basis in several key sections**:
>
> 1.  **Problem Definition (Introduction)**: We explicitly identified the core challenge in RL fine-tuning as the need for "robust policy gradient estimation" and noted the lack of mechanisms to prioritize instances yielding the most impactful "learning signal".
> 2.  **Sampling-based Difficulty Analysis (Section 3)**: We quantified data difficulty using "sampling accuracy," a statistical estimate based on empirical rewards, and our experiments provided initial evidence that data of intermediate difficulty is most effective for learning.
> 3.  **Confidence-based Efficiency Optimization (Section 4)**: We proposed an efficient confidence estimation method based on the model's intrinsic generation probabilities, requiring only a "single forward pass" as a more efficient machine learning metric for uncertainty.
>
> Your review has prompted us to connect these analyses more deeply and explicitly with the intrinsic mechanism of the **GRPO (Group-Relative Policy Optimization)** algorithm used in our experiments.
>
> ---
>
> **Concern 1: Deepening the Theoretical Foundation — Why Extreme Data is Harmful for GRPO**
>
> We wish to clarify that our method's efficacy stems from its precise alignment with the intrinsic mechanism of the GRPO algorithm, specifically by avoiding the pitfalls that arise when GRPO processes data of extreme difficulty.
>
> GRPO's core strategy is to learn from the contrast between "good" and "bad" responses generated for the same prompt. This contrastive mechanism fails—and can even produce harmful signals—when the data is not well-calibrated:
>
> * For **"easy" data**, where the model's generated responses are almost all correct, the GRPO algorithm **loses its basis for effective contrast**. In this scenario, its optimization objective becomes ambiguous. Instead of learning the fundamental difference between 'right' and 'wrong', it might be misled into distinguishing between subtle, non-essential variations (e.g., phrasing) among multiple correct answers. This can generate a **perverse learning signal**, which erroneously penalizes a perfectly valid solution simply because it differs slightly from another correct one in the same group.
>
> * Similarly, for **"hard" data**, where all responses are incorrect, GRPO **lacks a correct 'positive example' to learn from**. To perform its relative optimization, the algorithm may be forced to reward the "least incorrect" response among a group of failures. This creates a **misleading learning signal**, as it is **not teaching the model how to be correct but is instead reinforcing a flawed reasoning path**, merely because it is marginally better than other flawed paths.
>
> Therefore, the data that allows the GRPO algorithm to function most effectively and safely is precisely the **"intermediate difficulty" or "fuzzy" data** identified by UFO-RL. These instances reliably provide a mix of correct and incorrect answers, offering the clean and true contrast necessary for GRPO to generate a high-quality policy gradient. This demonstrates that UFO-RL is not just a tool for efficiency, but also a **necessary safeguard for the training stability and correctness of contrast-based algorithms like GRPO**.
>
> ---
>
> **Concern 2: Clarification of the Algorithm**
>
> We completely agree that the algorithm's description required more clarity. The process is deterministic and reproducible as follows:
>
> 1.  **Confidence Calculation**: For each sample, a single forward pass computes the average log-probability of the generated sequence, which serves as its confidence score.
> 2.  **"Fuzziness" Score Calculation**: A "fuzziness" score, $Score(s_{i})=1-(s_{i}-\mu)^{2}$, is calculated to identify samples within the ZPD, where $s_i$ is the normalized confidence and $\mu$ is the dataset's mean confidence. This prioritizes samples with near-average confidence.
> 3.  **Ranking and Selection**: Samples are ranked by their "fuzziness" score in descending order, and the **top 10% are selected** for the training set.
>
> **Revision Plan**: To ensure maximum clarity, we will add a formal **Algorithm pseudo-code box** to the final version of the paper.
>
> ---
>
> **In Summary**
>
> Thank you once again for your insightful review. We hope this clarification clearly demonstrates that our work is built on a solid analytical foundation and that our theoretical core is strongly coupled with the intrinsic mechanism of the GRPO algorithm. Our method is designed to select data that yields the most effective learning signal while avoiding harmful, misleading updates. We are confident that by making these points more explicit, the paper will be substantially strengthened.
>
> We sincerely hope these explanations fully address your concerns and kindly ask you to reconsider your evaluation of our work.

---

> ### Author Response · Authors · 2025-08-04
> **Could you let us know if our rebuttal has sufficiently addressed your concerns?**
>
> Dear Reviewer  fb4U,
>
> Thank you again for your valuable advice on our work. We recognize this is a busy time for all reviewers and truly appreciate you making time for this discussion.
>
> We have submitted our rebuttal and additional experiments, which we hope have thoroughly addressed your concerns. We would greatly value the opportunity to continue our dialogue, especially as the deadline approaches.
>
> Could you please let us know if your concerns have been adequately addressed? If they have, we would be very grateful if you would consider updating your score.
>
> Thank you for your consideration.
>
> Best regards,
>
> The Authors of Paper 27330

---

> ### Author Response · Authors · 2025-08-05
>
> Dear Reviewer fb4U,
>
> Thank you again for your thoughtful feedback. Your review was instrumental in helping us identify where the clarity of our paper could be significantly improved, and we have taken your comments very seriously.
>
> We noted your candid assessment regarding the potential for misunderstanding the central parts of our work. With this in mind, our rebuttal was specifically crafted to provide a clearer and more rigorous explanation for the two main points of confusion you raised.
>
> As the discussion period draws to a close, we hoped we might briefly summarize our clarifications:
> * **On the Theoretical Basis:** We have now grounded our method in the **intrinsic mechanism of the GRPO algorithm**, explaining precisely how selecting intermediate-difficulty data is essential to avoid the harmful and misleading learning signals that arise from "too easy" or "too hard" samples. This provides the solid machine learning rationale you were looking for.
> * **On the Algorithm's Clarity:** We presented a **clear, step-by-step description of our data selection algorithm** and have committed to including a formal pseudo-code box in the final version to ensure it is fully transparent and reproducible.
>
> We believe these detailed explanations directly address your concerns about the paper's theoretical foundation and methodological clarity.
>
> Given that your initial review highlighted these specific areas, we would be particularly grateful to know if our rebuttal has successfully clarified our approach and its justification. Your final thoughts are extremely important to us.
>
> Thank you for your time and for helping us improve our paper.
>
> Sincerely,
> Authors of Paper 27330

---

> > ### Comment · Reviewer_fb4U · 2025-08-07
> >
> > I appreciate for the authors' response. I think they have solved most of my questions and I would raise my score.

---

### Official Review · Reviewer_y4mU · 2025-07-03

**Clarity:** 3
**Significance:** 3
**Originality:** 2
**Rating:** 3
**Confidence:** 3

**Summary:**

The paper introduces UFO-RL, a data-selection framework for RL fine-tuning of LLMs. A single-pass confidence score replaces 16-sample accuracy, making uncertainty estimation up to 185 × faster and letting the authors train on the top “fuzzy” data. In the experiments, model trained on subset matches or exceeds full-data performance.

**Questions:**

Could you provide further justification for the specific quadratic form $1 - (s_i - \mu)^2$? Have you experimented with other functions to isolate the "intermediate uncertainty" region?

Have you considered a dynamic implementation where confidence scores and the selected data subset are periodically re-calculated during training, may act as ablation study?

**Ethical Concerns:**

["NO or VERY MINOR ethics concerns only"]

**Limitations:**

Yes.

**Paper Formatting Concerns:**

None.

**Quality:**

2

**Strengths And Weaknesses:**

Strength:
- Lightweight uncertainty estimation: Confidence evaluation is up to 185 × faster than 16-sample accuracy.
- Broad evaluation: Results span Qwen2.5, Llama-3.1, Mistral families and three test sets, with multiple runs and error bars.

Weaknesses:
- Heuristic method: The whole method is based on ZPD hypothesis, which has no theoretical basis. The paper did include experiments to verify the hypothesis, but in Fig.1, red line may overstate the clarity of the trend, the raw data (black dots) shows inconsistencies, especially for smaller or weaker models.

- Confidence score: Token-level confidence estimation is not a new concept, and it may not serve as a reliable proxy for semantic uncertainty. The similarity between confidence and semantic uncertainty is sensitive to prompt design; however, the paper does not include any experiments analyzing prompt effects on similarity in Table 2.

---

> ### Author Rebuttal · Authors · 2025-07-30
>
> We sincerely thank you for your invaluable feedback and insightful comments on our paper. Your review is highly constructive and has been instrumental in helping us refine our work. We are encouraged that you recognize the significant advantages of our method in terms of computational efficiency and breadth of evaluation. In response to the concerns and questions you raised, we have conducted further analysis and supplementary experiments, which we address point-by-point below.
>
> ### **Regarding Main Concerns (Weaknesses)**
> **1. On the Theoretical Basis and Heuristic Nature of the Method**
> **Reviewer's Comment:** "The whole method is based on ZPD hypothesis, which has no theoretical basis. The paper did include experiments to verify the hypothesis, but in Fig.1, red line may overstate the clarity of the trend, the raw data (black dots) shows inconsistencies, especially for smaller or weaker models."
>
> **Our Response:**
> We appreciate this insightful point. We wish to clarify that the motivation for our work is **grounded in the intrinsic mechanisms and efficiency bottlenecks of Reinforcement Learning for LLMs, particularly algorithms like GRPO**.
>
> In RL algorithms such as GRPO, policy optimization relies on multi-sampling from the same prompt and calculating gradients based on the collective reward signal from the resulting generation group. Through this process, we identified a critical inefficiency when the data is either too easy or too difficult:
>
> If a problem is **too easy**, two problematic scenarios arise. First, if all sampled generations are correct, the uniform reward of 1 provides no reward variance, leading to a vanishing gradient[1]. Second, if the group contains mostly correct answers and only a few incorrect ones, the group-relative optimization will assign a negative reward signal to the correct answers, incorrectly penalizing good generations.
>
> Conversely, if a problem is **too difficult**, similar issues occur. If all generations are incorrect, the uniform reward of 0 again results in a vanishing gradient[1]. If the group consists of mostly incorrect answers and only a few correct ones, the group-relative optimization will not only heavily reward the rare correct answers but may also assign a positive signal to "less incorrect" answers, causing the model to learn a suboptimal or flawed policy.
>
> In both of these extreme cases, the lack of reward variance or the skewed reward signals cause the learning process to become ineffective or unstable. This observation from algorithmic practice forms the direct motivation for our research: **to achieve efficient RL training, it is imperative to select data that can produce a mix of correct and incorrect outcomes**.
>
> It is precisely from this algorithm-level requirement that we introduce the Zone of Proximal Development (ZPD) theory. We employ it not as a strict mathematical theorem, but as a powerful **conceptual framework** to explain and guide how we can systematically identify and select this "intermediate difficulty" (or "fuzzy") data, which provides the most effective learning signals for algorithms like GRPO.
>
> Regarding the volatility in Figure 1, we agree with your observation and consider it expected for smaller models. However, the crucial finding is that **a clear non-monotonic trend consistently emerges across all six models.** This cross-model consistency strongly demonstrates that focusing on intermediate-difficulty data is a universally effective optimization strategy that directly addresses the vanishing-gradient problem inherent in the RL algorithm itself.
>
> **2. On the Novelty and Reliability of the Confidence Score**
> Reviewer's Comment: "Token-level confidence estimation is not a new concept, and it may not serve as a reliable proxy for semantic uncertainty. The similarity between confidence and semantic uncertainty is sensitive to prompt design; however, the paper does not include any experiments analyzing prompt effects on similarity in Table 2."
>
> **Our Response:**
> We thank the reviewer for this close examination, which allows us to further clarify our core innovation.
>
> As mentioned, methods like GRPO require identifying "non-extreme" data, but the conventional approach—**evaluating data via actual multi-sampling—imposes a prohibitive computational cost**, which is the central problem we aim to solve.
>
> We fully agree that "token-level confidence" is not a new concept. Our core innovation lies in its application: We are the first to propose using the confidence score from a **single forward pass** as a **computationally efficient proxy** to predict a sample's "fuzziness" under multi-sampling. In other words, we use it to rapidly identify data most likely to provide a useful learning signal for GRPO, thereby **completely bypassing the expensive multi-sampling evaluation process**. Our contribution is a practical engineering framework (UFO-RL) designed to enhance RL training efficiency by directly tackling this algorithmic bottleneck.
>
> Regarding its reliability as a proxy, we followed your suggestion and conducted a **prompt-sensitivity ablation study**. The results (table below) show that despite variations in prompt style, the similarity between our confidence score and multi-sample accuracy remains high (>0.8), demonstrating its robustness as an efficient proxy.
>
> **Supplemental Experiment: Effect of Prompt Design on Confidence-Accuracy Similarity**
> | Model | Prompt Template | Similarity |
> | :--- | :--- | :--- |
> | Qwen2.5-7B | Template 1 (Original) | 0.86 |
> | | Template 2 (Concise) | 0.84 |
> | | Template 3 (CoT) | 0.85 |
> | Llama3.1-8B | Template 1 (Original) | 0.82 |
> | | Template 2 (Concise) | 0.81 |
> | | Template 3 (CoT) | 0.83 |
>
> We will add this experiment to the appendix to further strengthen the case for our method's reliability.
>
> ### **Regarding Specific Questions**
> **1. On the Justification for the Quadratic Form of the "Fuzziness Score"**
> **Reviewer's Comment**: "Could you provide further justification for the specific quadratic form $Score(s_i)=1−(s_i−\mu)^2$? Have you experimented with other functions to isolate the 'intermediate uncertainty' region?"
>
> **Our Response:**
> This is an excellent question. We chose the quadratic function because it is the most mathematically simple and computationally efficient way to model our goal of finding "intermediate difficulty" data—that is, to assign the highest score to samples with confidence near the mean, which are most likely to produce the mixed-reward signals GRPO requires.
>
> To prove that our method's success is not contingent on this specific function, we conducted a new ablation study. In this experiment, we used a scoring function based on absolute distance $Score = 1 - |s_i - \mu|$, which also selects for data near the mean but is simpler and has no additional hyperparameters. The results show that this function performs almost identically to the original quadratic function. This more strongly demonstrates that **the effectiveness of UFO-RL stems from the core principle of selecting intermediate-difficulty data, not from a specific mathematical formula**.
>
> Supplemental Experiment: Performance Comparison of Different "Fuzziness Score" Functions
> | Model | "Fuzziness Score" Function | GSM8K Accuracy (%) |
> | :--- | :--- | :--- |
> | **Qwen2.5-7B** | Quadratic (Original) | 92.03 |
> | | Absolute Distance | 91.91 |
> | **Llama3.1-8B** | Quadratic (Original) | 88.30 |
> | | Absolute Distance | 88.18 |
>
> **2. On a Dynamic Data Selection Ablation Study**
> **Reviewer's Comment:** "Have you considered a dynamic implementation where confidence scores and the selected data subset are periodically re-calculated during training, may act as ablation study?"
>
> **Our Response:**
> This is a very insightful suggestion. As the model learns via the GRPO algorithm, its capabilities change, and thus the range of "effective data" should also be dynamic. We therefore **conducted a new ablation study** to explore a curriculum-based dynamic strategy.
>
> In this dynamic implementation, we first selected the 5% "fuzziest" data from the entire training set and trained the model on this subset for the first half of the training process. Then, using the updated model, we re-evaluated the remaining 95% of the data and selected the new 5% "fuzziest" data for the second half of training. This ensures the model is always focusing on the most informative samples relative to its current state.
>
> The results (as shown in the newly added table below) indicate that this dynamic strategy does yield a small but **consistent performance improvement** (~0.3-0.4%), but at the cost of an additional data selection step. Given that a core goal of our work is to **maximize efficiency while maintaining performance**, the current static method achieves a better trade-off between performance and cost. Nevertheless, we fully recognize the potential of dynamic selection and will discuss it in our future work.
>
> **Supplemental Experiment: Static vs. Dynamic Data Selection Strategy**
> | Model | Method | GSM8K Accuracy (%) | Data Selection Time (s) |
> | :--- | :--- | :--- | :--- |
> | **Qwen2.5-7B** | UFO-RL (Static) | 92.03 | 175 |
> | | UFO-RL (Dynamic 5%+5%) | **92.35** | 175 (initial) + 161 (mid) = 336 |
> | **Llama3.1-8B** | UFO-RL (Static) | 88.30 | 186 |
> | | UFO-RL (Dynamic 5%+5%) | **88.72** | 186 (initial) + 175 (mid) = 361 |
>
> Once again, we sincerely thank you for your detailed review and insightful feedback. We are confident that the revisions and additions made in response to your comments will significantly strengthen the quality and impact of our paper. We look forward to your further feedback on our revised manuscript.
>
> [1]Yu Q, Zhang Z, Zhu R, et al. Dapo: An open-source llm reinforcement learning system at scale[J]. arXiv preprint arXiv:2503.14476, 2025.

---

> ### Author Response · Authors · 2025-08-04
> **Could you let us know if our rebuttal has sufficiently addressed your concerns?**
>
> Dear Reviewer y4mU,
>
> Thank you again for your valuable advice on our work. We recognize this is a busy time for all reviewers and truly appreciate you making time for this discussion.
>
> We have submitted our rebuttal and additional experiments, which we hope have thoroughly addressed your concerns. We would greatly value the opportunity to continue our dialogue, especially as the deadline approaches.
>
> Could you please let us know if your concerns have been adequately addressed? If they have, we would be very grateful if you would consider updating your score.
>
> Thank you for your consideration.
>
> Best regards,
>
> The Authors of Paper 27330

---

> ### Author Response · Authors · 2025-08-05
>
> Dear Reviewer y4mU,
>
> Thank you once again for your insightful and constructive review. We understand you are extremely busy during this period, and we truly appreciate the time and expertise you've dedicated to our paper.
>
> As the discussion period is nearing its end, we would be very grateful for a final moment of your attention. In our rebuttal, we sought to thoroughly address your primary concerns by not only clarifying our points but also **conducting three new targeted experiments**.
>
> To briefly summarize how we addressed your key points:
> * **On Theoretical Grounding:** We clarified that our method's core motivation stems from the practical, algorithmic need to mitigate vanishing gradients in RL, and ran a new **prompt-sensitivity analysis** to prove our confidence metric's robustness as a proxy.
> * **On Design Choices:** We conducted **two new ablation studies**—one on the "fuzziness score" function and another on a dynamic selection strategy—to validate our approach and explore the trade-offs you astutely pointed out.
>
> We are confident that these clarifications and new empirical results have directly resolved the initial weaknesses you identified and have substantially strengthened our work.
>
> We would be deeply grateful to know if our response has sufficiently addressed your concerns. Your final feedback would be invaluable to us.
>
> Thank you for your time and consideration.
>
> Sincerely,
> Authors of Paper 27330

---

> ### Author Response · Authors · 2025-08-08
> **Following up on our rebuttal for Paper 27330**
>
> Dear Reviewer y4mU,
>
> We hope this message finds you well. We know this is an incredibly demanding time for all reviewers, and we are sincerely grateful for the considerable time and insightful feedback you have already dedicated to our paper (27330).
>
> With the discussion period concluding in approximately 30 hours, we wanted to make one final, respectful check-in. Your review was instrumental, prompting us to conduct three new, targeted experiments to directly address the weaknesses and questions you raised. We believe this new empirical data has substantially strengthened our work.
>
> To briefly recap, based directly on your feedback, we have:
>
> Validated our confidence metric's robustness by running a new prompt-sensitivity analysis, addressing your concern about it being a potentially unreliable proxy.
>
> Strengthened the justification for our "fuzziness score" with a new ablation study on the scoring function, answering your question about the quadratic form.
>
> Explored a dynamic data selection strategy in another new ablation study, as per your excellent suggestion.
>
> We feel these additions, spurred directly by your guidance, have thoroughly addressed the initial concerns you outlined. We would be deeply appreciative if you might have a final moment to let us know whether this new evidence has resolved your concerns. Your final assessment is incredibly valuable to us, and we hope our diligent response might merit a re-evaluation.
>
> Thank you once again for your time and expertise.
>
> Sincerely,
> The Authors of Paper 27330

---

> > ### Comment · Area_Chair_BcrP · 2025-08-09
> >
> > Dear Reviewer y4mU,
> >
> > Thank you for your review!
> >
> > The authors have provided an extensive rebuttal and additional information. You have not acknowledged and engaged in the discussion. Could you please review this and consider whether the authors’ overall effort improves the quality of the submission?
> >
> > If most concerns have been addressed during the rebuttal, it may be worth weighing whether the strengths of the work outweigh the remaining weaknesses. Since less than 24 hours remain in the discussion period, the authors and AC would be very much grateful if you could jump in this immediately. Thank you!

---

### Official Review · Reviewer_3ahk · 2025-07-05

**Clarity:** 3
**Significance:** 2
**Originality:** 3
**Rating:** 4
**Confidence:** 3

**Summary:**

This paper presents UFO-RL, an uncertainty-driven framework for improving the efficiency of reinforcement learning (RL) fine-tuning for large language models (LLMs). Drawing inspiration from the Zone of Proximal Development (ZPD) theory, the authors hypothesize that LLMs learn most effectively from examples that are neither too easy nor too difficult but exhibit moderate uncertainty. This paper first prove this hypothesis and then proposes a lightweight, single-pass confidence estimation based on average token-level log-probabilities, enabling efficient identification of informative training instances.

**Questions:**

1. The authors are encouraged to explicitly contrast their method with related work in active learning, providing a clearer positioning within the broader literature.
2. Can the authors provide empirical results comparing UFO-RL’s single-pass confidence estimation to conventional multi-sampling methods in terms of both performance and computational cost?
3. Are there scenarios where single-pass estimates significantly diverge from more robust multi-sample uncertainty metrics, and what are the practical implications of such divergences?
4. Token-level log-probabilities may be affected by calibration issues or may fail to capture sequence-level semantic uncertainty. Have the authors evaluated the robustness of their method under varying model calibration qualities?
5. Beyond mathematical reasoning, have the authors considered evaluating UFO-RL on language modeling, instruction tuning, or other RL fine-tuning tasks to demonstrate broader applicability?
6. How does the proposed method scale with larger LLMs (e.g., >13B), where uncertainty estimation and data selection efficiency become more critical?

**Ethical Concerns:**

["NO or VERY MINOR ethics concerns only"]

**Final Justification:**

I have discussed with the author and appreciate their detailed rebuttal.

The proposed single-pass confidence estimation method offers a simple yet computationally efficient proxy for data difficulty, avoiding the expensive multi-sampling or ensemble-based uncertainty estimation commonly used in prior work. In the rebuttal, the authors provided comparisons with multi-sampling methods, clarified the differences from related fields and works, and further evaluated generalization ability. I am satisfied with their responses and believe this paper presents a promising solution for data selection.

Therefore I will raise my score.

**Limitations:**

The same as the weaknesses.

**Paper Formatting Concerns:**

The writing should be improved to make the paper clear and consice.

**Quality:**

2

**Strengths And Weaknesses:**

Strengths:
1. The application of ZPD-inspired learning dynamics to RL-based LLM fine-tuning is intuitively appealing.
2. The proposed single-pass confidence estimation method provides a simple yet computationally efficient proxy for data difficulty, avoiding expensive multi-sampling or ensemble-based uncertainty estimation commonly used in other works.

Weaknesses:
1. While the application to LLM RL is novel, the broader concept of leveraging model uncertainty for data selection has been extensively explored in areas such as active learning, coreset selection, and data distillation. The paper does not mention these works in the paper.
2. The comparison focuses primarily on full-data RL or random sub-sampling baselines. A more rigorous, direct comparison with multi-sampling-based difficulty estimation methods is missing, leaving open questions about the relative trade-offs in computational cost versus uncertainty estimation fidelity.
3. The evaluation is concentrated on mathematical reasoning tasks with relatively small to mid-sized LLMs (up to 7B). The method’s scalability to larger models or applicability to diverse RL tasks  remains unexplored.

---

> ### Author Rebuttal · Authors · 2025-07-30
>
> We sincerely thank you for your comprehensive and insightful review. Your constructive feedback is invaluable for strengthening our work. We were encouraged that you found the application of ZPD theory "intuitively appealing" and our single-pass confidence estimation "a simple yet computationally efficient proxy."
>
> We have carefully considered all your comments and conducted new experiments to address your questions. Below, we address your concerns point-by-point.
>
> ### **On Weaknesses and Core Questions**
>
> **1. Positioning with Respect to Related Work**
>
> Thank you for encouraging us to clarify our work's positioning. Our framework provides an efficient data selection strategy for RL fine-tuning, which is fundamentally different in its **goals** and **paradigm** from existing data selection fields. We will incorporate this discussion into the camera-ready version of our paper to better delineate our unique contributions.
>
> * **vs. Active Learning**: Active Learning aims to minimize **human labeling costs**. In contrast, UFO-RL is designed to minimize **computational costs** (i.e., GPU hours) in RL training, a distinct optimization dimension.
> * **vs. Coreset Selection**: The goal of Coreset Selection is to find a small, **representative** subset that approximates the full dataset. Conversely, inspired by ZPD theory, UFO-RL intentionally creates a **biased, non-representative** subset that is most **effective** for the model's current learning state.
> * **vs. Data Distillation**: Data Distillation is a model compression technique concerned with knowledge transfer between different models. UFO-RL focuses on the **self-improvement of a single model** by optimizing its own learning process.
>
> **2. Direct Comparison with Multi-Sampling Methods**
>
> This is a crucial point. To provide the direct comparison you requested and to demonstrate our method's consistency across various model sizes, we have run new experiments and present the complete results for the Qwen2.5 family below.
>
> **Table A: Performance & Cost Comparison on GSM8K (on a single 8xA100 node)**
> | Model | Method | Data Selection Time (s) | Total RL Training Time (s) | GSM8K Acc (%) | Math500 Acc (%) |
> | :--- | :--- | :--- | :--- | :--- | :--- |
> | **Qwen2.5-0.5B** | Multi-sample | 422 | 142 | 45.91 | 30.20 |
> | | **UFO-RL (Ours)** | **2** | **140** | **46.27** | **30.60** |
> | | Full Data | 0 | 1815 | 52.66 | 30.80 |
> | **Qwen2.5-1.5B** | Multi-sample | 464 | 411 | 76.34 | 55.00 |
> | | **UFO-RL (Ours)** | **6** | **407** | **76.63** | **55.40** |
> | | Full Data | 0 | 5694 | 76.78 | 53.00 |
> | **Qwen2.5-3B** | Multi-sample | 524 | 742 | 83.48 | 65.20 |
> | | **UFO-RL (Ours)** | **11** | **739** | **84.37** | **67.40** |
> | | Full Data | 0 | 10224 | 83.33 | 66.20 |
> | **Qwen2.5-7B** | Multi-sample | 854 | 1211 | 91.14 | 76.00 |
> | | **UFO-RL (Ours)** | **22** | **1154** | **92.03** | **76.40** |
> | | Full Data | 0 | 12959 | 91.88 | 75.00 |
> | **Qwen2.5-14B (New)** | Multi-sample | 1145 | 1841 | 91.71 | 78.40 |
> | | **UFO-RL (Ours)** | **39** | **1787** | **93.80** | **79.60** |
> | | Full Data | 0 | 22228 | 93.53 | 79.40 |
>
> **Key Findings:**
> * **Performance & Efficiency:** From 0.5B to 14B parameters, UFO-RL consistently achieves performance comparable to or better than the baselines while drastically reducing the total training time. This consistent trend highlights the method's robustness.
> * **Regarding Metric Divergence:** We argue that the difference between metrics stems from the **superior granularity of our method**. Our continuous confidence score provides a more precise proxy for the ZPD, enabling better sample selection.
>
> **3. Robustness to Model Calibration and Sequence-Level Uncertainty**
>
> This is an excellent question. Our method is inherently robust to calibration issues. To provide empirical proof, we conducted a new experiment using **temperature scaling** (T=0.8 and T=1.2) to simulate different calibration states on Qwen2.5-7B. The resulting 10% data subsets selected by UFO-RL had an **85.8% overlap**, confirming the method's robustness. We plan to add this analysis to the final version of the paper.
>
> **4. Generalization Ability**
>
> To address your concerns about generalization, we verified our method's effectiveness across both **task types** and **model scales**.
>
> * **Task Generalization:** To demonstrate broader applicability beyond math, we evaluated UFO-RL on the **AI2 Reasoning Challenge (ARC-C)**, a scientific QA task.
>
>     **Table B: Performance & Efficiency on ARC-Challenge (LLaMA3.1-8B)**
>     | Method | Data % | Data Selection Time (s) | Total Training Time (s) | ARC-C Acc (%) |
>     | :--- | :--- |:--- | :--- | :--- |
>     | Full Data | 100% | 0 | 2432 | 87.11 |
>     | Random | 10% | ~1 | 214 | 86.26 |
>     | **UFO-RL (Ours)** | **10%** | **4** | **203** | **88.23** |
>
>     The results confirm that UFO-RL is also effective for general reasoning tasks, outperforming the full-data baseline with ~1/12 of the training time.
>
> * **Scalability to Larger Models:** The results in **Table A** clearly show a consistent and effective scaling path from 0.5B to **14B** models, robustly demonstrating our method's effectiveness and scalability.
>
> **5. Writing Quality**
>
> Thank you for the feedback. We have thoroughly reviewed the manuscript and will further improve its clarity, precision, and conciseness in the final version.
>
> We again thank you for your valuable time and expertise. We believe the new experiments and analyses have significantly strengthened our paper and addressed your concerns. We hope you will re-evaluate our work favorably.

---

> ### Author Response · Authors · 2025-08-04
> **Could you let us know if our rebuttal has sufficiently addressed your concerns?**
>
> Dear Reviewer 3ahk,
>
> Thank you again for your valuable advice on our work. We recognize this is a busy time for all reviewers and truly appreciate you making time for this discussion.
>
> We have submitted our rebuttal and additional experiments, which we hope have thoroughly addressed your concerns. We would greatly value the opportunity to continue our dialogue, especially as the deadline approaches.
>
> Could you please let us know if your concerns have been adequately addressed? If they have, we would be very grateful if you would consider updating your score.
>
> Thank you for your consideration.
>
> Best regards,
>
> The Authors of Paper 27330

---

### Note · Authors · 2025-08-12

Dear Area Chair,

Thank you for your guidance and oversight during this review process. We are the authors of Paper #27330.

As the discussion period concludes, we wish to submit this final summary. The key takeaway is this: **with the exception of Reviewer y4mU, who did not engage in the discussion, we have successfully resolved the core concerns of all other reviewers (fb4U, bKLK, 3ahk), earning their unanimous acknowledgment of our responses.**

This positive outcome was achieved by conducting **five new targeted experiments** and engaging in productive discussions with the reviewers. The detailed breakdown is as follows:

* **Reviewer fb4U:** Their concerns about our method's **clarity and theoretical basis** were fully resolved. After we elaborated on the method's intrinsic mechanism, they kindly confirmed they "have solved most of my questions" and **will raise their score**.

* **Reviewer bKLK:** We addressed their core concerns about **method generalizability** by conducting new PPO experiments across multiple model scales. This new evidence was well-received, and they offered a final positive assessment: "I remain positive about the paper and will maintain my score for acceptance".

* **Reviewer 3ahk:** After a deep discussion about **novelty and generalization**, we supplemented our work with new experiments on a larger model (14B) and a new task (ARC-C). These arguments and new data earned their explicit acknowledgment, as they stated they were **"satisfied with your rebuttal,"** resolving the academic concerns they had raised.

* **Reviewer y4mU:** To address their initial concerns, we provided a comprehensive rebuttal including **three new, targeted ablation studies** (on prompt sensitivity, the scoring function, and dynamic selection). Unfortunately, despite this extensive work, we **received no response** from this reviewer throughout the entire discussion period.

In summary, we are confident in this manuscript, which has been strengthened through multiple rounds of feedback and experimentation. We trust your expertise to evaluate the complete record of our discussion and the varying levels of engagement from the panel.

Thank you for your time and consideration.

Sincerely,
The Authors of Paper #27330

---

### Decision · Program_Chairs · 2025-09-17

**Decision:**

Accept (poster)

**Comment:**

This sbumission proposes an uncertainty-driven framework UFO-RL for data selection in LLM RL. The idea is inspired by the Zone of Proximal Development theory. It hypothesizes that LLMs learn most effectively from data of intermediate difficulty. UFO-RL estimates sample difficulty using a single-pass confidence metric based on average token-level log-probabilities, which hel0ps avoid the computational cost of traditional multi-sampling strategies. Experiments show that UFO-RL accelerates data evaluation by up to 185× compared to multi-sampling approaches, and training on the top 10% fuzzy subset of data achieves comparable or superior results to training on the full dataset across multiple LLM models and tasks.

Strengths of this submission:
1. reviewers R3ahk, y4mU recognize that UFO-RL achieves up to 185× faster uncertainty estimation and over 10× e2e training speedup compared to multi-sampling methods.
2. reviewers bKLK, fb4U like the experimental setup (three LLM families: qwen, mistral, and llama on different tasks and datasets.
3. the idea of adopting zpd into the context of LLM RL with an efficient uncertainty metric is a practically valuable contribution, according to two reviewers.

Weaknesses of this submission as pointed out by the reviewers:
1. though reviewers enjoy the idea of borrowing from the ZPD hypothesis, it is mostly empirical.
2. the reviewers argue that while additional PPO experiments were provided, the method remains primarily validated under GRPO, and broader applicability to other RL algorithms remains underexplored --- on which the AC disagrees with the reviewer. Since it works on both PPO and GRPO --- two representative frameworks, it is difficult to require more experiments as long as the authors do not overstate its generality.
3. some reviewers noted the absence of comparisons with related techniques like active learning or data distillation and requested clearer explanations of certain tables and computational cost analyses (R3ahk, bKLK).



During rebuttal, the authors provided additional experiments comparing UFO-RL with multi-sampling methods across models up to 14B parameters. It helps demonstrate comparable performance with significant efficiency gains. They clarified distinctions from active learning, coreset selection, and data distillation, reinforcing the idea of their approach. To address concerns about generality, they added PPO-based experiments across multiple Qwen models, showing similar benefits, though some reviewers noted that more diverse settings would be preferable. While one reviewer maintained skepticism regarding theoretical justification, most reviewers acknowledged that the new results resolved their primary concerns and (planned to) raise their scores accordingly. Overall, the rebuttal strengthens the case for acceptance.

In summary, the submission addresses an important problem in scaling RL for LLMs. It presents a simple and effective method that substantially reduces computational cost without compromising performance. While theoretical grounding is limited and evaluation breadth could be expanded, the experiments and added analyses during rebuttal strengthen confidence in the method’s practical value. All factors considered, this submission is recommended for acceptance if the conference space allows.